# Interleukin-4 induces CD11c+ microglia leading to amelioration of neuropathic pain in mice

**Keita Kohno[1], Ryoji Shirasaka[1], Keita Hirose[1], Takahiro Masuda[2], Makoto Tsuda[1,3]***

[1]Department of Molecular and System Pharmacology, Graduate School of Pharmaceutical Sciences, Kyushu University, Fukuoka, Japan; [2]Division of Molecular Neuroimmunology, Medical Institute of Bioregulation, Kyushu University, Fukuoka, Japan; [3]Kyushu University Institute for Advanced Study, Fukuoka, Japan

## eLife Assessment

This study is **important** as it highlighted how IL-4 regulates the reactive state of a specific microglial population by increasing the proportion of CD11c+ microglial cells and ultimately suppressing neuropathic pain. The study employs a combination of behavioral assays, pharmacogenetic manipulation of microglial populations, and characterization of microglial markers to address these questions. It provided **convincing** evidence for the proposed mechanism of IL-4-mediated microglial regulation in neuropathic pain.

*For correspondence:
tsuda@phar.kyushu-u.ac.jp

**Abstract** Neuropathic pain, a debilitating chronic pain condition, is a major clinical challenge. The pleiotropic cytokine interleukin-4 (IL-4) has been shown to suppress neuropathic pain in rodent models, but its underlying mechanism remains unclear. Here, we show that intrathecal administration of IL-4 to mice with spinal nerve transection (SpNT) increased the number of CD11c+ microglia (a microglia subset important for pain remission) in the spinal dorsal horn (SDH) and that this effect of IL-4 was essential for its ameliorating effect on SpNT-induced pain hypersensitivity. Furthermore, in mice with spared nerve injury (SNI), another model in which pain remission does not occur, the emergence of CD11c+ SDH microglia was curtailed, but intrathecal IL-4 increased their emergence and ameliorated pain hypersensitivity in a CD11c+ microglia-dependent manner. Our study reveals a mechanism by which intrathecal IL-4 ameliorates pain hypersensitivity after nerve injury and provides evidence that IL-4 increases CD11c+ microglia with a function that ameliorates neuropathic pain.

## Introduction

Neuropathic pain is a debilitating chronic pain condition that results from injury or disease of the somatosensory system. Since chronic pain is refractory to currently available treatments, the development of effective treatments is a major clinical challenge (*Colloca et al., 2017*; *Finnerup et al., 2021*). Accumulating evidence from studies using models of neuropathic pain indicates that peripheral nerve injury causes various inflammatory responses in the nervous system that critically contribute to the development of neuropathic pain (*Ji et al., 2016*; *Inoue and Tsuda, 2018*). Especially, pro-inflammatory cytokines [e.g. interleukin-1β (IL-1β), IL-6 and tumor necrosis factor α (TNFα)] produced by various non-neuronal cells such as macrophages at injured nerves and dorsal root ganglion (DRG), and glial cells (mainly microglia and astrocytes) in the spinal dorsal horn (SDH) modulate function of primary afferent sensory neurons and SDH neurons, leading to an increase in the excitability of pain signaling neural pathway (*Ji et al., 2016*; *Inoue and Tsuda, 2018*; *Ji et al., 2019*; *Grace et al., 2014*).

Along with these findings, much attention has been paid to the pain-relieving role of anti-inflammatory cytokines (*Vanderwall and Milligan, 2019*). These include IL-10, the role of which in chronic pain has been extensively studied (*Vanderwall and Milligan, 2019*). Another anti-inflammatory cytokine with pain-relieving effects in diverse models, including neuropathic pain, is IL-4. Intraplantar or systemic administration of IL-4 has been shown to attenuate pain behavioral responses to bradykinin (*Cunha et al., 1999*), carrageenin (*Cunha et al., 1999*), TNFα (*Cunha et al., 1999*), zymosan (*Vale et al., 2003*), and acetic acid (*Vale et al., 2003*) in normal animals. In models of neuropathic pain, single or repeated administration of IL-4 around the injured sciatic nerve reduced behavioral pain hypersensitivity (*Celik et al., 2020*; *Kiguchi et al., 2015*; *Labuz et al., 2021*). These effects of IL-4 have been proposed to involve a decrease in the production of inflammatory mediators (*Kiguchi et al., 2015*), polarization of macrophages toward an anti-inflammatory phenotype at injured nerves (*Celik et al., 2020*; *Kiguchi et al., 2015*), increases in the production and release of IL-10 (*Kiguchi et al., 2015*) and opioid peptides (*Celik et al., 2020*; *Labuz et al., 2021*) from macrophages, and inhibition of the increased action potentials of DRG neurons (*Nie et al., 2018*). In addition to the diverse actions of IL-4 at the peripheral level, the central nervous system, including the spinal cord, is recognized as an important locus of action of IL-4 in pain control. Intrathecal administration of IL-4 prevents the development of mechanical pain hypersensitivity in several neuropathic pain models (*Nie et al., 2018*; *Okutani et al., 2018*). The target cells for the action of IL-4 intrathecally administered could be microglia since, within the SDH of nerve-injured rats, IL-4 receptor α chain (IL-4Rα) is predominantly expressed in microglia (*Okutani et al., 2018*). In line with this, intrathecal IL-4 activates signal transducer and activator of transcription 6 (STAT6; an intracellular signal transduction molecule downstream of IL-4Rα) in SDH microglia (*Okutani et al., 2018*). Intrathecal IL-4 or other manipulation to increase spinal IL-4 during the early phase after nerve injury suppresses cellular indexes for reactive microglia (*Okutani et al., 2018*), IL-1β and Prostaglandin E2 release, and microglial p38 phosphorylation (*Hao et al., 2006*), all of which are required to develop pain hypersensitivity (*Inoue and Tsuda, 2018*). In addition to the effect of IL-4 on pain development, it is noteworthy that intrathecal administration of IL-4 during the late phase (e.g. 2 weeks or later after nerve injury) leads to recovery from a behavioral pain-hypersensitive state that develops following nerve injury (*Okutani et al., 2018*). Despite the remarkable therapeutic potential of spinal IL-4 for ameliorating established neuropathic pain, little is known about its mechanism of action. Given that the reactive states of microglia occur early after nerve injury (e.g. cell number and expression of proinflammatory genes) and subside in the late phase (*Inoue and Tsuda, 2018*; *Kohno et al., 2018*; *Tansley et al., 2022*), it is possible that the pain-resolving effect of IL-4 involves a mechanism other than the previously reported suppression of inflammatory responses in activated microglia.

In this study, using two different models of neuropathic pain, we demonstrated that intrathecally administered IL-4 during the late phase changes microglia to a CD11c[+] state, which has recently been shown to be necessary for the spontaneous remission of behavioral hypersensitivity associated with neuropathic pain (*Kohno et al., 2022*), and, importantly, that this is required for the pain-relieving effect of intrathecal IL-4. Consequently, our findings uncover a mechanism for the ameliorating effect of spinal IL-4 on already-developed neuropathic pain and provide evidence at a preclinical level that IL-4 could induce the emergence of CD11c[+] microglia with a function that resolves neuropathic pain.

## Results

### Intrathecal treatment of IL-4 increases CD11c[+] microglia in the SDH and alleviates pain hypersensitivity after SpNT

To examine the effect of IL-4 on the behavioral remission of neuropathic pain, we used mice with spinal nerve transection (SpNT; a model of neuropathic pain *Ho Kim and Mo Chung, 1992*; *Rigaud et al., 2008*) and repeated the administration of IL-4 to their intrathecal spaces from day 14 post-SpNT, when the paw withdrawal threshold (PWT) decreased (indicating that mechanical pain hypersensitivity had been developed) (*Figure 1A*). Intrathecal treatment with IL-4 increased the PWT in the hindpaw ipsilateral (but not contralateral) to the SpNT (*Figure 1A*). Notably, this behavioral remission of neuropathic mechanical hypersensitivity persisted for at least 5 days after the last intrathecal injection of IL-4. Because we previously showed that CD11c[+] microglia emerging in the SDH after SpNT are necessary for the spontaneous remission of pain behavior that gradually occurs around 3 weeks

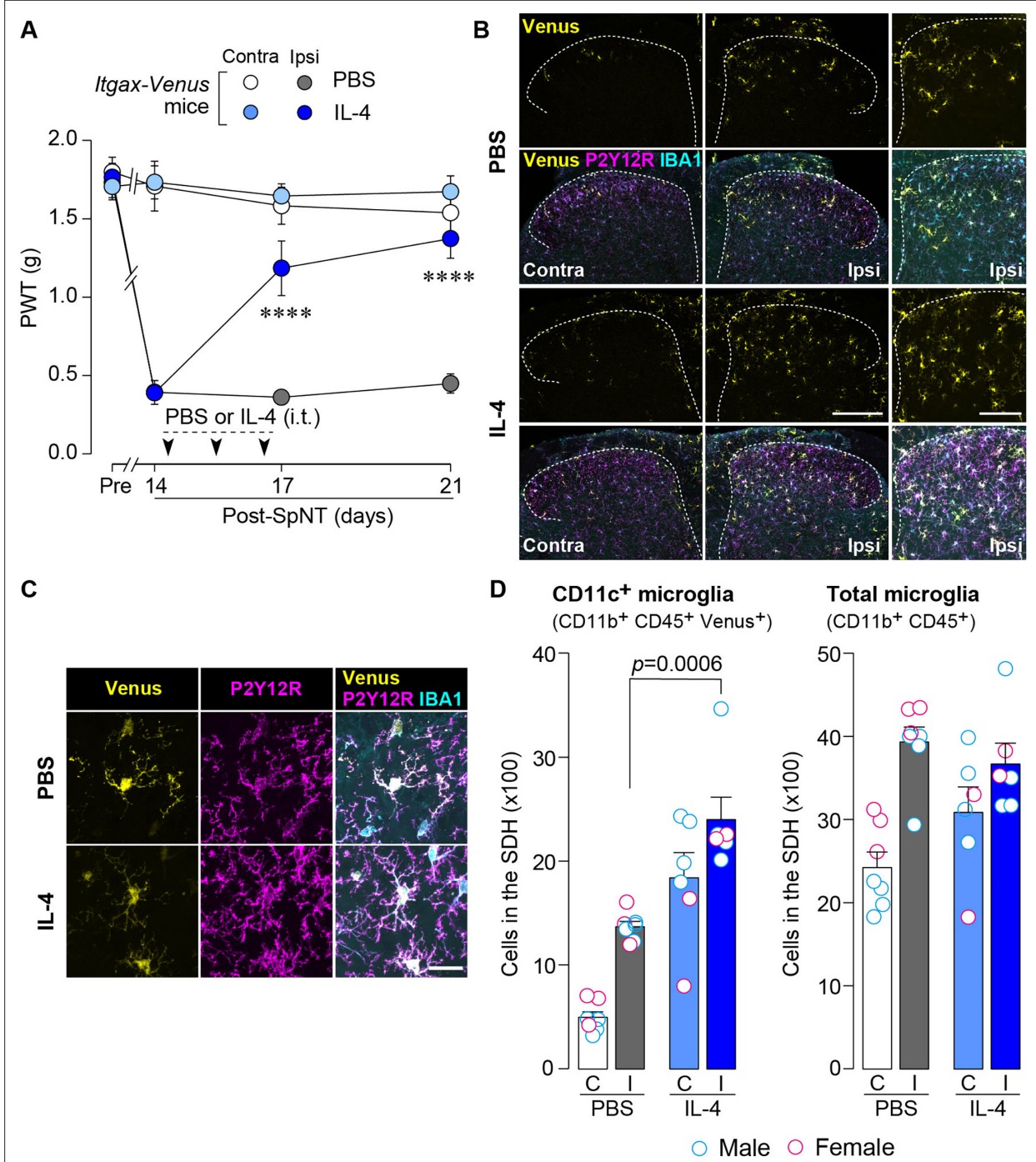

**Figure 1.** IL-4 increases CD11c+ microglia in the SDH and ameliorates pain hypersensitivity in the SpNT mice. (**A**) Paw withdrawal threshold (PWT) of *Itgax-Venus* mice before (Pre) and after SpNT (n=7–8 mice). IL-4 or PBS was intrathecally administrated from day 14 to day 16 post-SpNT (once a day for 3 days). ****p<0.0001 versus the ipsilateral side of PBS-treated group, two-way ANOVA with post hoc Tukey multiple comparison test. (**B**) Venus fluorescence (yellow) and P2Y12R and IBA1 immunostaining (magenta and cyan, respectively) in the SDH of *Itgax-Venus* mice with SpNT 21 days after PBS or IL-4 treatment (from day 14–16). Scale bars, 200 μm (middle), 100 μm (right). (**C**) Colocalization of Venus, P2Y12R, and IBA1. Scale bar, 20 μm. (**D**) Flow cytometric quantification for the number of CD11c+ (CD11b+CD45+Venus+) and total (CD11b+CD45+) microglia in the 3–4th lumbar SDH contralateral (**C**) and ipsilateral (**I**) to SpNT (n=6–7 mice). One-way ANOVA with post hoc Tukey multiple comparison test. Data are shown as mean ± SEM.

The online version of this article includes the following source data and figure supplement(s) for figure 1:

**Source data 1.** Raw numerical values for *Figure 1* plots.

**Figure supplement 1.** Monocytes/macrophages in the SDH and DRG after intrathecal administration of IL-4.

after nerve injury (*Kohno et al., 2022*), we explored the mechanism of action of spinal IL-4 with a focus on CD11c+ microglia in the SDH. In *Itgax-Venus* mice, in which CD11c+ cells are visualized using the fluorescent protein Venus (*Kohno et al., 2022*; *Lindquist et al., 2004*), SpNT increased the number of Venus+ (CD11c+) cells in the SDH ipsilateral to the injury (*Figure 1B*). Consistent with our previous data (*Kohno et al., 2022*), almost all CD11c+ cells were immunohistochemically positive for ionized calcium-binding adapter molecule 1 (IBA1) and the purinergic P2Y12 receptor (P2Y12R; *Figure 1B and C*), which are markers for myeloid cells and microglia, respectively (*Butovsky et al., 2014*; *Ito et al., 1998*). We found that intrathecal administration of IL-4 to SpNT mice enhanced the increase in CD11c+ cells in the SDH (*Figure 1B–D*). These cells also expressed both IBA1 and P2Y12R (*Figure 1B and C*), indicating that CD11c+ cells increased by intrathecal IL-4 were microglia. Quantitative flow cytometry analysis revealed a significant increase in CD11c+ microglia in the SDH of IL-4-treated SpNT mice (*Figure 1D*). In contrast, IL-4 did not significantly change the total number of microglia in the SDH (*Figure 1D*). In addition, intrathecal IL-4 did not induce infiltration of CD169+ monocytes/macrophages into the SDH (*Figure 1—figure supplement 1A*). These findings suggest that CD11c+ microglia may be involved in the mechanism by which intrathecal treatment with IL-4 attenuates pain hypersensitivity.

## IL-4-induced remission of pain hypersensitivity requires spinal CD11c+ microglia

To determine whether CD11c+ microglia are required for pain remission induced by IL-4, we used *Itgax-DTR-EGFP* mice in which the diphtheria toxin (DTX) receptor (DTR) and enhanced green fluorescence protein (EGFP) are expressed on CD11c+ cells (*Kohno et al., 2022*; *Jung et al., 2002*), and administered DTX intrathecally to these mice to deplete CD11c+ microglia in the spinal cord.

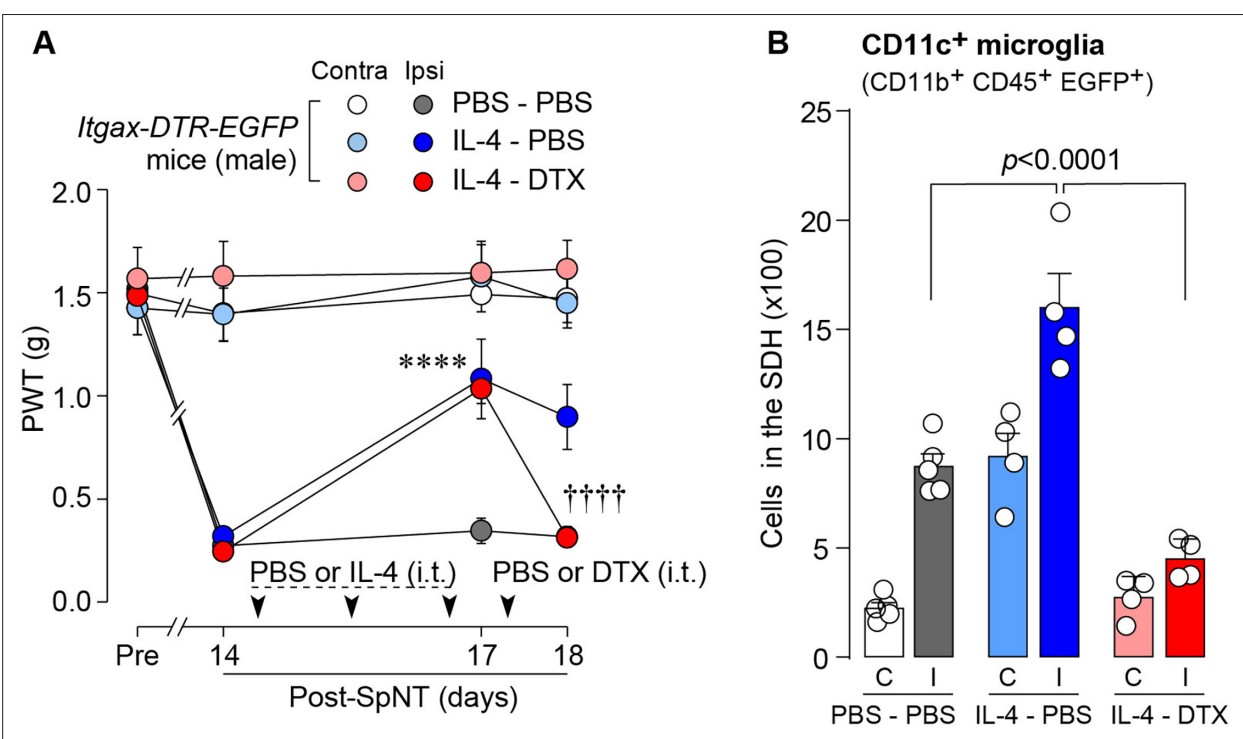

**Figure 2.** IL-4-induced remission of pain hypersensitivity requires spinal CD11c+ microglia. (**A**) Paw withdrawal threshold (PWT) of *Itgax-DTR-EGFP* mice before (Pre) and after SpNT (n=6–7 mice). IL-4 or PBS was intrathecally administrated from days 14–16 post-SpNT (once a day for 3 days). On day 17, DTX (0.5 ng/mouse) or PBS was intrathecally injected. ****p<0.0001 versus the ipsilateral side of PBS-PBS group, ††††p<0.0001 versus the ipsilateral side of IL-4-PBS group, two-way ANOVA with post hoc Tukey multiple comparison test. (**B**) Flow cytometric quantification for the number of CD11c+ microglia (CD11b+CD45+EGFP+) in the 3–4th lumbar SDH contralateral (**C**) and ipsilateral (**I**) to SpNT in *Itgax-DTR-EGFP* mice treated with PBS/PBS, IL-4/PBS, or IL-4/DTX (n=4–5 mice). One-way ANOVA with post hoc Tukey multiple comparison test. Data are shown as mean ± SEM.

The online version of this article includes the following source data for figure 2:

**Source data 1.** Raw numerical values for *Figure 2* plots.

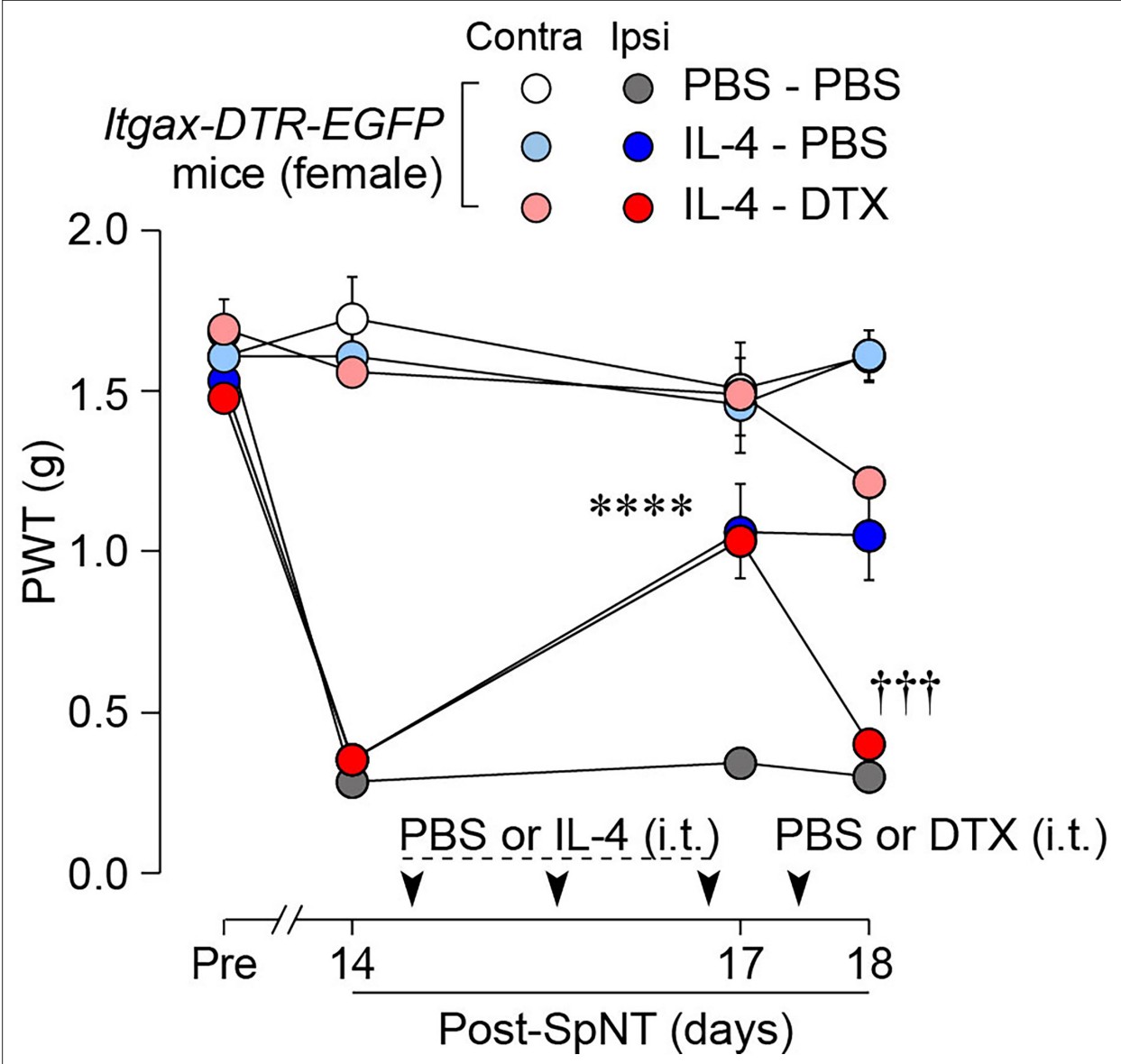

**Figure 3.** IL-4 alleviates pain hypersensitivity in female mice. Paw withdrawal threshold (PWT) of *Itgax-DTR-EGFP* mice before (Pre) and after SpNT (n=6 female mice). IL-4 or PBS was intrathecally administered from days 14–16 post-SpNT (once a day for 3 days). On day 17, DTX (0.5 ng/mouse) or PBS was intrathecally injected. ****p<0.0001 versus the ipsilateral side of PBS-PBS group, †††p<0.001 versus the ipsilateral side of IL-4-PBS group, two-way ANOVA with post hoc Tukey multiple comparison test. Data are shown as mean ± SEM.

The online version of this article includes the following source data for figure 3:

**Source data 1.** Raw numerical values for *Figure 3* plots.

Intrathecal IL-4 also induced behavioral remission of mechanical hypersensitivity in *Itgax-DTR-EGFP* mice, but the pain-reducing effect was eliminated by intrathecal injection of DTX on day 17 post-SpNT (*Figure 2A*). The depletion of EGFP+ (CD11c+) microglia in the SDH by DTX injection was confirmed (*Figure 2B*), indicating the necessity of spinal CD11c+ microglia for the pain-relieving effect of intrathecal IL-4. IL-4 also alleviated pain hypersensitivity in female mice, which was also dependent on CD11c+ microglia (*Figure 3*). In addition to SDH, CD11c+ cells have been shown to be present at the site of injury and in the DRG (*Kohno et al., 2022*). Thus, we examined the involvement of CD11c+ cells in the DRG in the pain-relieving effect of intrathecal IL-4 using our previously reported strategy that enables the selective depletion of CD11c+ cells in the periphery (*Kohno et al., 2022*). The intraperitoneal administration of DTX at a low dose (2 ng/g mouse) in SpNT mice treated with IL-4 on day 17

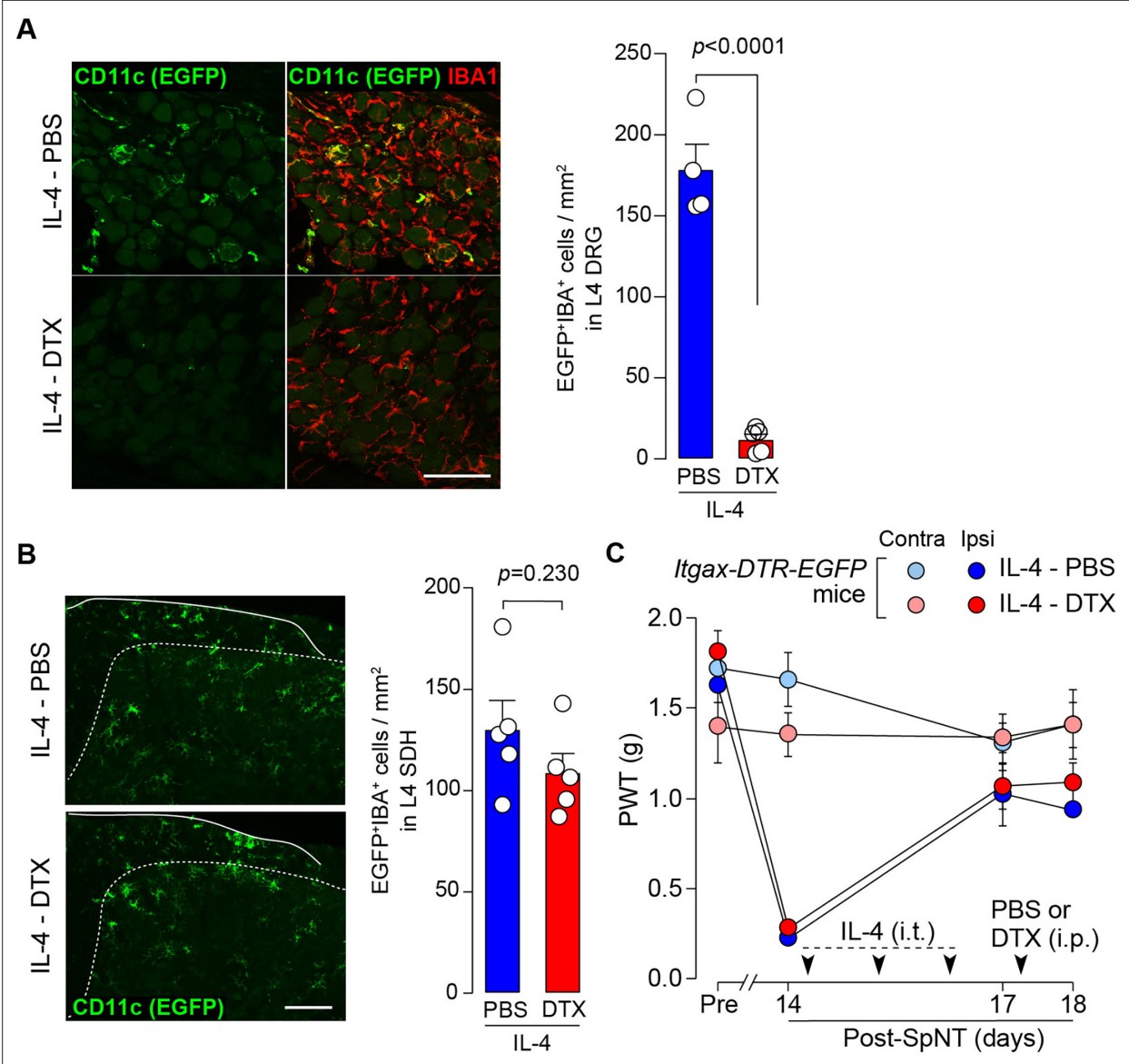

**Figure 4.** CD11c+ cells in the DRG are involved in IL-4-induced remission of pain hypersensitivity. (**A** and **B**) Immunohistochemical analyses of CD11c+ cells (EGFP+ IBA1+) in the DRG (**A**) and SDH (**B**) on day 18 post-SpNT of *Itgax-DTR-EGFP* mice treated intrathecally with IL-4 (from days 14 to 16) (n=4–5 mice). DTX (2 ng/g) or PBS was intraperitoneally injected on day 17. The myeloid marker IBA1 was also stained in the SDH (**B**). Scale bars, 100 μm. Unpaired t-test. (**C**) Effect of DTX (2 ng/g) intraperitoneally injected (on day 17) on PWT of SpNT mice treated intrathecally with PBS or IL-4 (from days 14 to 16) (n=5 mice). Data are shown as mean ± SEM.

The online version of this article includes the following source data for figure 4:

**Source data 1.** Raw numerical values for **Figure 4** plots.

efficiently eliminated CD11c+ cells (also positive for IBA1) in the DRG on day 18 (**Figure 4A**). However, the number of CD11c+ microglia in the SDH was not significantly different between the PBS- and DTX-treated groups (**Figure 4B**). Under these conditions, we found that the elevated PWT observed in IL-4-treated mice on day 17 did not change 1 day after intraperitoneal administration of DTX (day 18; **Figure 4C**), suggesting that CD11c+ cells in the DRG are not involved in the IL-4-induced alleviating effect on neuropathic pain. Altogether, these data indicate that intrathecally administered IL-4 induces spinal microglia to be in a CD11c+ state, and that this induction is necessary for the ameliorating effect of IL-4 on neuropathic pain.

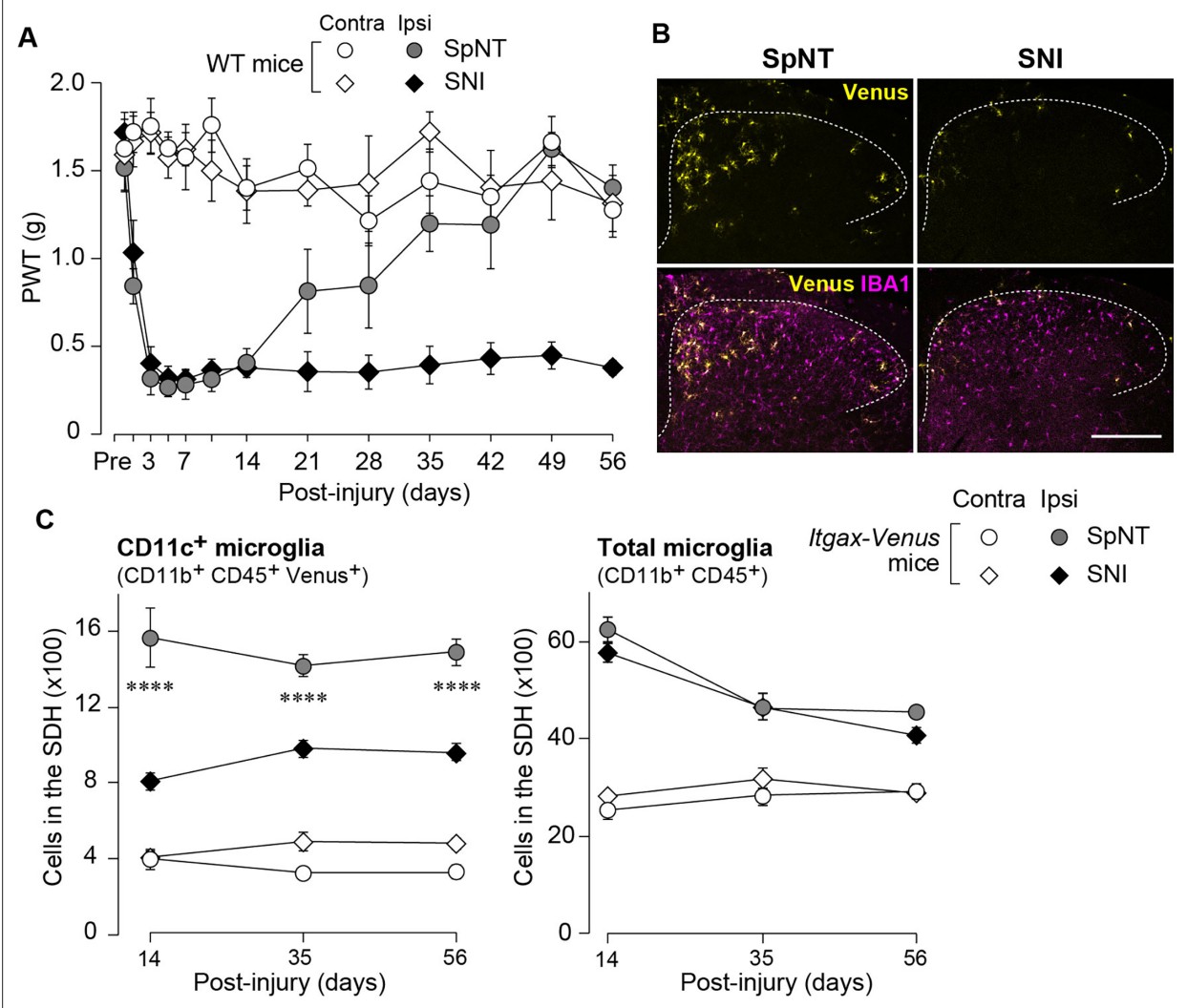

**Figure 5.** Blunted appearance of CD11c+ microglia in the SDH after SNI. (**A**) PWT of wild-type (WT) mice before (Pre) and after SpNT and SNI (n=5 mice). (**B**) Venus (yellow) and IBA1 immunostaining (magenta) in the SDH of SpNT and SNI mice on day 14. Scale bar, 200 μm. (**C**) Flow cytometric quantification of the number of CD11c+ (CD11b+CD45+Venus+) and total (CD11b+CD45+) microglia in the 3–4th lumbar SDH of *Itgax-Venus* mice after SpNT and SNI (n=3–5 mice). ****p<0.0001 versus the ipsilateral side of the SNI group, two-way ANOVA with post hoc Tukey multiple comparison test. Data are shown as mean ± SEM.

The online version of this article includes the following source data for figure 5:

**Source data 1.** Raw numerical values for *Figure 5* plots.

## CD11c+ microglia appearance in the SDH is blunted in SNI model

To extend the therapeutic potential of CD11c+ microglia induced by IL-4 in neuropathic pain, we employed another model of neuropathic pain [speared nerve injury (SNI); transection of tibial and common peroneal nerves] (*Decosterd and Woolf, 2000*). Consistent with previous data (*Decosterd and Woolf, 2000*), SNI induced a long-lasting pain-hypersensitive state without spontaneous remission, at least until the last day of testing (day 56 post-SNI; *Figure 5A*), which was in stark contrast to the time course of pain hypersensitivity in the SpNT model. We found that the number of CD11c+ microglia in the SDH at 14 days after nerve injury was lower in the SNI model than in the SpNT model (*Figure 5B*). Quantitative analysis by flow cytometry confirmed that there were fewer CD11c+ SDH microglia in the SNI mice (*Figure 5C*). A blunted appearance was also observed on days 35 and 56 post-SNI. In contrast, the total number of microglia in the SDH was comparable between the two models at all time points tested, suggesting an impairment of the nerve injury-induced appearance of CD11c+ microglia in the SNI model. Furthermore, insulin-like growth factor 1 (IGF1) has been

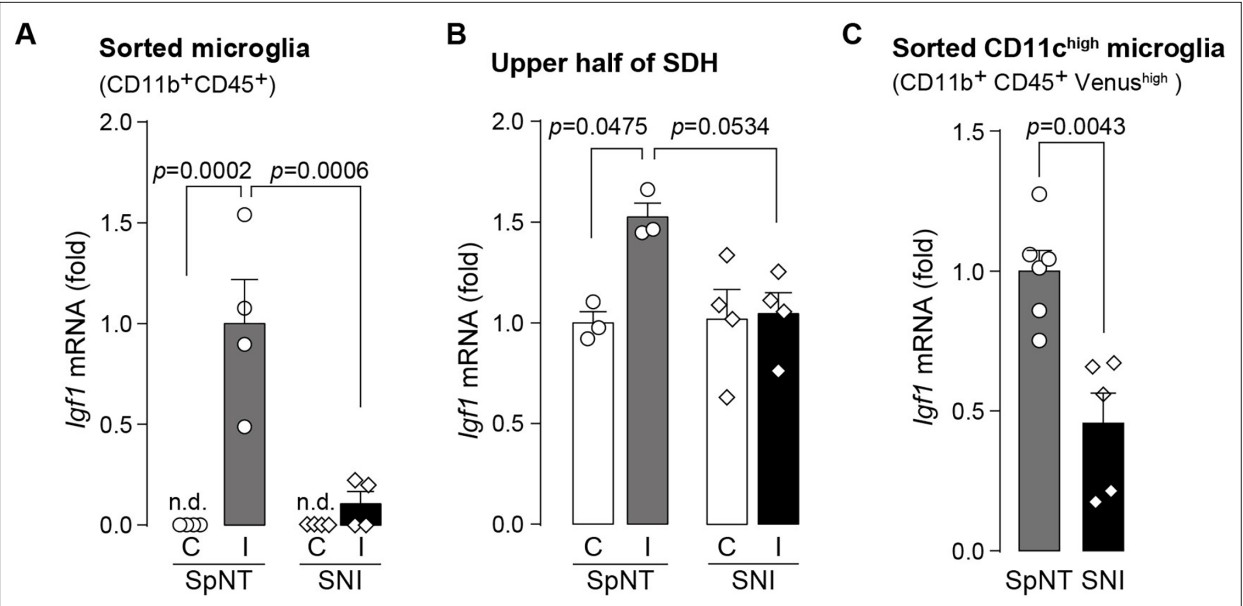

**Figure 6.** Expression of Igf1 is lower in the SNI model. (**A–C**) *Igf1* mRNA in total RNA extracted from sorted SDH microglia (*A*; n=4 mice), from tissue homogenate of the upper half of the SDH (*B*; n=3–4 mice), or from the sorted CD11chigh microglia (*C*; n=5–6 mice) in *Itgax-Venus* mice was quantified by qPCR on day 36 post-SpNT or SNI. Cells and tissues were collected from the contralateral (**C**) and ipsilateral (**I**) side to the SpNT or SNI. Values represent the relative ratio of *Igf1* mRNA (normalized to the value for *Actb* mRNA) to the ipsilateral (**A and C**) or contralateral (**B**) side of SpNT mice. One-way ANOVA with post hoc Tukey multiple comparison test (**A and B**). Unpaired t-test (**C**). Data are shown as mean ± SEM.

The online version of this article includes the following source data for figure 6:

**Source data 1.** Raw numerical values for *Figure 6* plots.

identified as a factor that is highly expressed in CD11c+ microglia and is necessary for their pain-remitting effect (*Kohno et al., 2022*). We quantified *Igf1* mRNA expression in the spinal tissue or fluorescence-activated cell sorting (FACS)-isolated SDH microglia from nerve-injured mice (*Figure 6*). While *Igf1* expression in sorted microglia increased in both models after injury, its levels were lower in microglia from SNI mice than in those from SpNT mice (*Figure 6D*). In addition, an increase in *Igf1* mRNA expression in the tissue homogenate of the upper half of the SDH, where CD11c+ microglia accumulate (*Kohno et al., 2022*), was not observed in tissues from the SNI model (*Figure 6E*). Moreover, in FACS-isolated CD11chigh SDH microglia that highly expressed IGF1, the expression of *Igf1* mRNA was significantly lower in the SNI model (*Figure 6F*). Thus, given that either the depletion of CD11c+ microglia or knockout of their IGF1 prevents the spontaneous remission of pain in the SpNT model (*Kohno et al., 2022*), the long-lasting pain-hypersensitive state in the SNI model could be related to the blunted increase in CD11c+ microglia in the SDH and their lower expression of IGF1.

## IL-4 alleviates SNI-induced pain hypersensitivity in CD11c+ microglia- and IGF1-dependent manners

Based on the above findings, we tested whether IL-4 could increase the number of CD11c+ microglia and resolve pain hypersensitivity in the SNI model. Intrathecal treatment with IL-4 in SNI mice from day 14 to day 16 significantly increased the number of CD11c+ microglia on day 21, almost all of which were positive for IBA1 and P2Y12R (*Figure 7A–C*). Similar to the data obtained for SpNT mice, the total number of microglia in the SDH was not significantly different between PBS- and IL-4-treated SNI mice (*Figure 7C*). Behaviorally, intrathecal IL-4 treatment alleviated the SNI-induced mechanical hypersensitivity (*Figure 7D*). Furthermore, the depletion of CD11c+ microglia in the SDH of IL-4-treated SNI mice by intrathecal administration of DTX on day 17 negated the ameliorating effect of IL-4 on pain hypersensitivity (*Figure 8A and B*). Moreover, the IL-4's effect was not observed in *Cx3cr1*CreERT2/+;*Igf1*flox/flox mice treated with tamoxifen (*Figure 8C*), a treatment that enables selective knockout of the *Igf1* gene in tissue-resident *Cx3cr1*+ cells (*Parkhurst et al., 2013*), the majority of

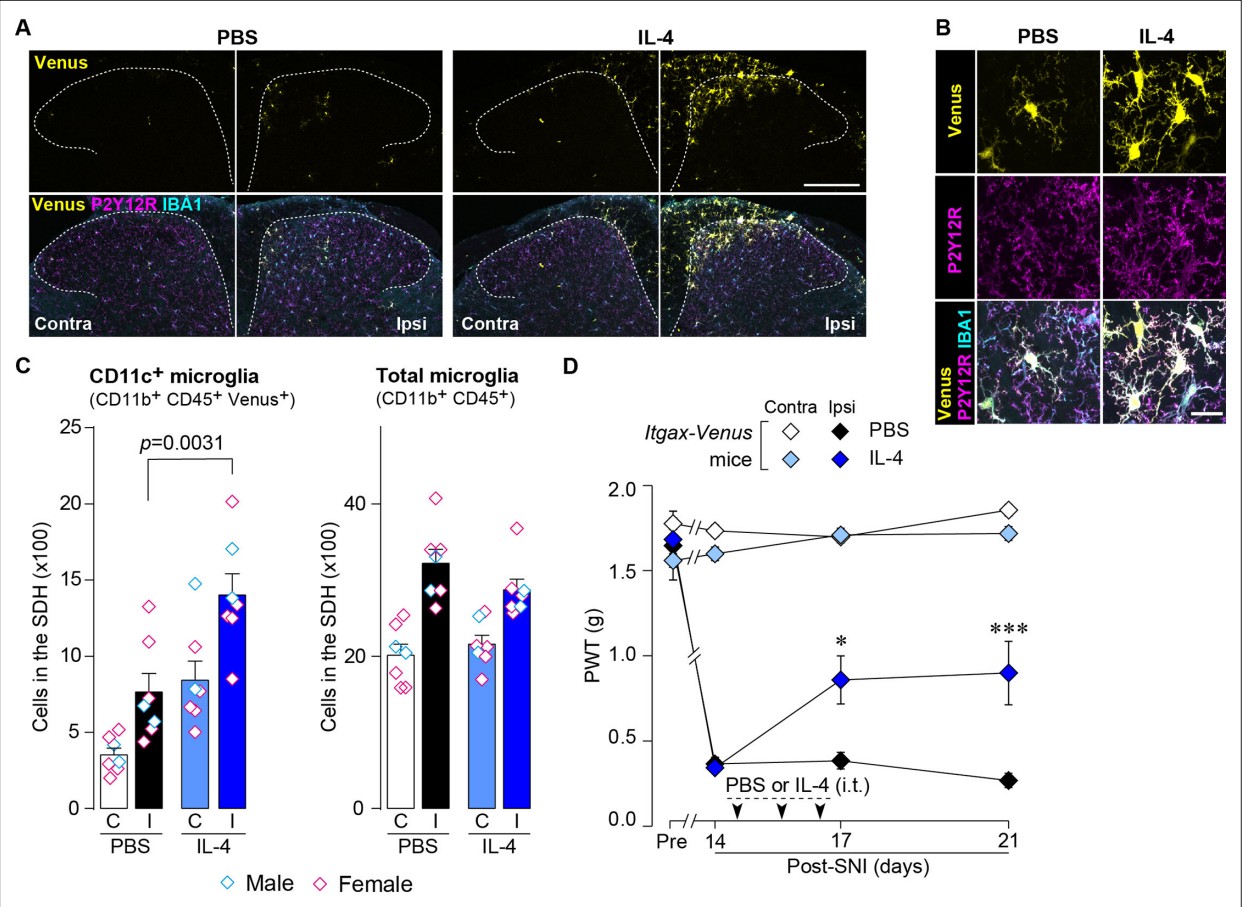

**Figure 7.** IL-4 increases CD11c+ microglia in the SDH and ameliorates pain hypersensitivity in the SNI mice. (**A**) Venus fluorescence (yellow) and P2Y12R and IBA1 immunostaining (magenta and cyan, respectively) in the SDH of *Itgax-Venus* mice with SpNT after PBS or IL-4 treatment. Scale bar, 200 μm. (**B**) Colocalization of Venus, P2Y12R, and IBA1. Scale bar, 20 μm. (**C**) Flow cytometric quantification for the number of CD11c+ (CD11b+CD45+Venus+) and total (CD11b+CD45+) microglia in the 3–4th lumbar SDH contralateral (**C**) and ipsilateral (**I**) to SNI (n=7 mice). One-way ANOVA with post hoc Tukey multiple comparison test. (**D**) Paw withdrawal threshold (PWT) of *Itgax-Venus* mice before (Pre) and after SNI (n=6–8 mice). IL-4 or PBS was intrathecally administrated from days 14 to 16 post-SNI (once a day for 3 days). *p<0.05, ***p<0.001 versus the ipsilateral side of PBS group, two-way ANOVA with post hoc Tukey multiple comparison test. Data are shown as mean ± SEM.

The online version of this article includes the following source data for figure 7:

**Source data 1.** Raw numerical values for *Figure 7* plots.

which are microglia in the SDH (*Kohno et al., 2022*). Overall, intrathecally administered IL-4 exhibited a pain-remitting effect in the SNI model, which was dependent on CD11c+ microglia and IGF1.

## Discussion

Increasing evidence indicates that following peripheral nerve injury, microglia in the SDH cause significant changes in morphology, cell number, transcriptional and translational activities, and function (*Inoue and Tsuda, 2018*; *Tansley et al., 2022*; *Shibata et al., 2025*; *Masuda et al., 2012*; *Masuda et al., 2014*). Numerous studies have shown that these changes occurring in the early phase after nerve injury are necessary for the development of neuropathic pain (*Inoue and Tsuda, 2018*; *Masuda et al., 2012*; *Masuda et al., 2014*; *Tsuda et al., 2003*; *Coull et al., 2005*). In contrast to their role, we have recently shown that some SDH microglia in the later phase are changed into CD11c+ states with different transcriptional and functional profiles and that these microglia are required for spontaneous remission of neuropathic pain (*Kohno et al., 2022*). Thus, CD11c+ microglia are attracting attention as a potential target for future analgesics (*Sideris-Lampretsas and Malcangio, 2022*; *Tsuda et al., 2023*). In this study, we demonstrated for the first time that CD11c+ microglia are an essential target

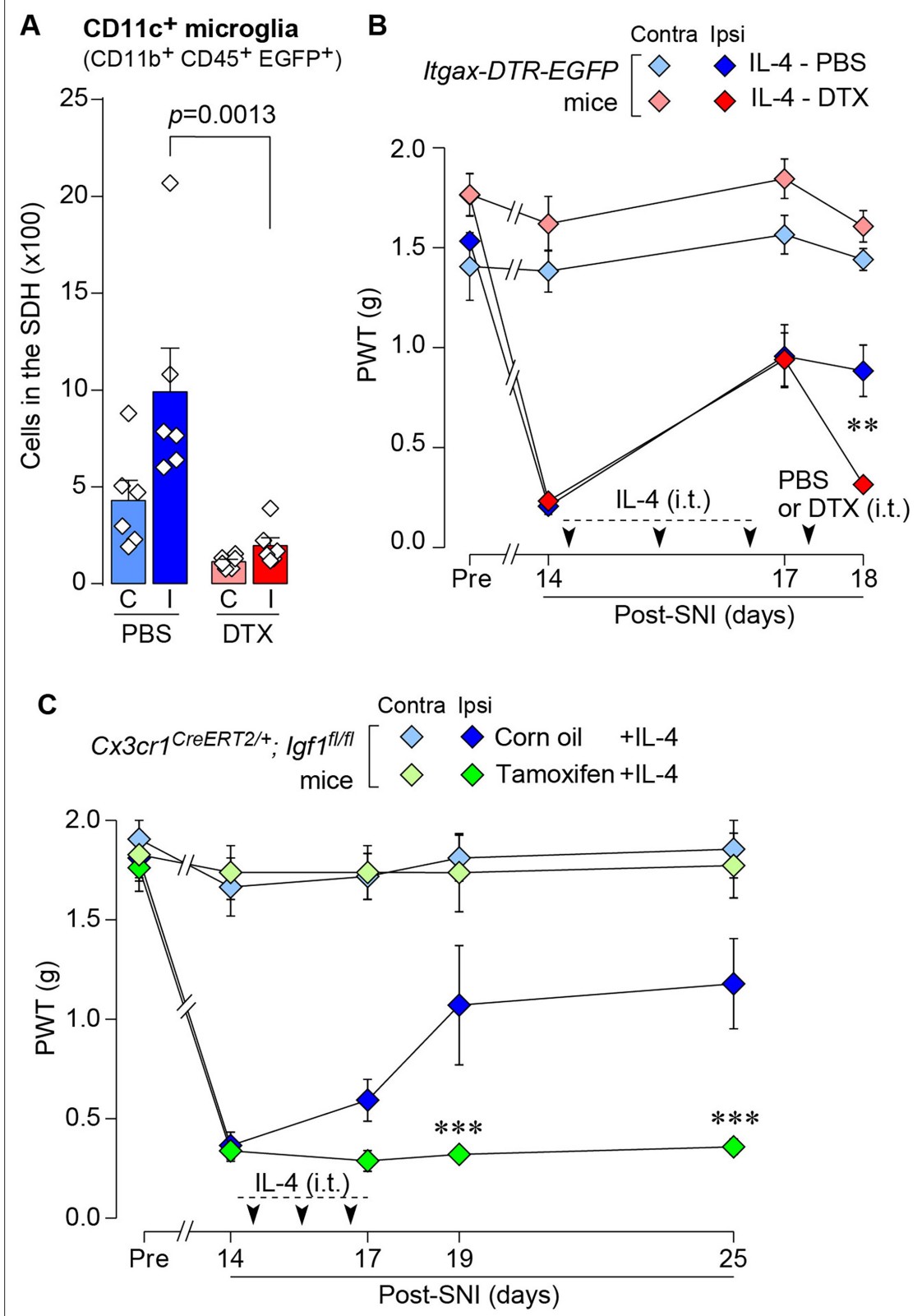

**Figure 8.** IL-4 alleviates SNI-induced pain hypersensitivity in CD11c⁺ microglia and IGF1-dependent manners. (**A**) Flow cytometric quantification for the number of CD11c⁺ microglia (CD11b⁺CD45⁺EGFP⁺) in the 3–4th lumbar SDH contralateral (**C**) and ipsilateral (**I**) to SNI in *Itgax-DTR-EGFP* mice treated with IL-4/PBS or IL-4/DTX (n=6 mice). One-way ANOVA with post hoc Tukey multiple comparison test. (**B**) PWT of *Itgax-DTR-EGFP* mice before (Pre) and after SNI (n=6 mice). IL-4 was intrathecally administered from days 14 to 16 post-SNI (once a day for 3 days). On day 17, DTX (0.5 ng/mouse) or PBS

*Figure 8 continued on next page*

*Figure 8 continued*

was intrathecally injected. **p<0.01 versus the ipsilateral side of the IL-4-PBS group, two-way ANOVA with post hoc Tukey multiple comparison test. (**C**) PWT of *Cx3cr1*^CreERT2/+;*Igf1*^flox/flox mice before (Pre) and after SNI (n=5–7 mice). Tamoxifen or vehicle was administered 4 weeks before SNI to induce recombination. IL-4 was intrathecally administered from days 14 to 16 post-SNI (once a day for 3 days). ***p<0.001 versus the ipsilateral side of Corn oil group, two-way ANOVA with post hoc Tukey multiple comparison test. Data are shown as mean ± SEM.

The online version of this article includes the following source data for figure 8:

**Source data 1.** Raw numerical values for *Figure 8* plots.

of IL-4 administered intrathecally to induce the behavioral remission of pain hypersensitivity. The pharmacological effect of intrathecal IL-4 appears to be mediated by its direct action on spinal microglia, as the expression of IL-4Rα in the SDH has been shown to be predominant in microglia after nerve injury (*Okutani et al., 2018*). This is also supported by the data that activation of STAT6 (the downstream molecule of IL-4Rα) after intrathecal IL-4 selectively occurs in spinal microglia (*Okutani et al., 2018*) and that primary cultured microglia treated with IL-4 become CD11c⁺ and express IGF1 (*Butovsky et al., 2006*; *Chiu et al., 2008*; *Butovsky et al., 2005*). Furthermore, based on our data showing that IL-4 increased the number of CD11c⁺ microglia without affecting the total number of microglia, it is conceivable that IL-4 acts on microglia and changes their state to CD11c⁺ microglia (rather than the proliferation of CD11c⁺ microglia themselves), leading to behavioral remission of neuropathic pain.

Intrathecal IL-4 administration also increased CD11c⁺ microglia in the SDH contralateral to SpNT but did not change the PWT in the contralateral hindpaw. This gap seems to be related to the selective effect of CD11c⁺ microglia and their factors (especially IGF1) on pathologically heightened pain sensitivity. In fact, depletion of CD11c⁺ spinal microglia and intrathecal administration of IGF1 do not elevate pain threshold of the contralateral hindpaw (*Kohno et al., 2022*).

In addition to SDH, the peripheral effects of IL-4 on neuropathic pain have been reported in several pain models. IL-4 applied to injured peripheral nerves, such as sciatic nerves, ameliorates neuropathic mechanical hypersensitivity (*Kiguchi et al., 2015*). In sciatic nerves, IL-4Rα expression is upregulated in macrophages that accumulate in injured nerves (*Kiguchi et al., 2015*). Intrathecally administered IL-4 may affect cells in the DRG, since chemokine (C-C motif) receptor 2 (CCR2)⁺ monocytes were increased in the capsule of the DRG after IL-4 treatment (*Figure 1—figure supplement 1B*). The involvement of these monocytes in the DRG in IL-4-induced alleviation of neuropathic pain is unclear and warrants further investigation. However, our findings obtained from mice with depletion of CD11c⁺ IBA1⁺ cells (presumably macrophages) in the DRG (but not SDH) suggest that the cells required for intrathecal IL-4 to ameliorate pain hypersensitivity are CD11c⁺ microglia in the SDH, but not peripheral CD11c⁺ macrophages. Furthermore, perineurally administered IL-4 can act on macrophages, which leads to a change in their state from pro- to anti-inflammatory via a STAT6-dependent mechanism (*Kiguchi et al., 2015*) and further to a release of opioid peptides that are key factors in the suppression of neuropathic pain (*Celik et al., 2020*). In the SDH, IGF1 appears to be necessary for the pain-relieving effect of CD11c⁺ microglia induced by IL-4. This is supported by data obtained from microglia-selective IGF1-knockout mice (*Cx3cr1*^CreERT2/+;*Igf1*^flox/flox mice treated with tamoxifen). Although *Cx3cr1*⁺ cells, other than CD11c⁺ microglia, also lack *Igf1* in these mice, given *Igf1* expression is much higher in CD11c⁺ microglia than in CD11c^neg microglia (*Kohno et al., 2022*), IGF1 from CD11c⁺ microglia would play an important role in IL-4-induced pain remission. Thus, while IL-4 acts on macrophages and microglia in the DRG/injured nerves and the SDH, respectively, the molecular basis of its ameliorating effects on neuropathic pain hypersensitivity is different.

Another notable finding of this study was that the emergence of CD11c⁺ spinal microglia following nerve injury was impaired in the SNI model, which did not exhibit spontaneous remission of behavioral pain hypersensitivity. This raises the possibility that the reduced ability of spinal microglia to shift toward the CD11c⁺ state in the SNI model contributes, at least in part, to long-lasting pain hypersensitivity because SpNT mice with depleted CD11c⁺ spinal microglia have been shown to exhibit persistent pain hypersensitivity (*Kohno et al., 2022*). Considering our finding that the total number of spinal microglia was comparable between the SpNT and SNI models, the signal(s) for microglia to transition to the CD11c⁺ state, rather than to the initial activation after nerve injury, could be different in these models. The mechanisms underlying this difference require further investigation, but we highlight that even under such conditions, IL-4 administered intrathecally induces the appearance of CD11c⁺ microglia in the SDH and exerts a pain-remitting effect in a manner that is dependent

on CD11c⁺ microglia. These data suggest that spinal microglia in the SNI model retain the ability to respond to IL-4, allowing microglia to be in a CD11c$^+$ state that remits neuropathic pain and broadly extends their therapeutic potential to neuropathic pain.

In this study, we showed an ability of IL-4, known as a T-cell-derived factor, to increase CD11c$^+$ microglia and to control neuropathic pain. Recent studies have also indicated that immune cells such as CD8$^+$ T cells infiltrating into spinal cord (*Fan et al., 2025*), and regulatory T cells (*Kuhn et al., 2021*; *Midavaine et al., 2025*) and MRC1$^+$ macrophages (*Niehaus et al., 2021*) in the spinal meninges have important roles in controlling microglial states and neuropathic pain. Thus, these findings provide new insights into the mechanisms of the neuro-immune interactions that regulate neuropathic pain. Furthermore, our data, demonstrating that IL-4 increases CD11c$^+$ microglia without affecting the total number of microglia, could open a new avenue for developing strategies to modulate microglial subpopulations through molecular targeting, which may lead to new analgesics. However, given IL-4's association with allergic responses, further investigation of the specific signaling pathways and molecular processes by which IL-4 induces a transition of spinal microglia to the CD11c$^+$ state may provide clues to discovering new targets that offer more selective and safer therapeutic approaches. Moreover, since CD11c$^+$ microglia have been implicated in other CNS diseases (e.g. Alzheimer disease *Keren-Shaul et al., 2017*, amyotrophic lateral sclerosis *Xie et al., 2022*, and multiple sclerosis *Wlodarczyk et al., 2018*), further investigations into the mechanisms driving CD11c$^+$ microglia induction could provide insights into novel therapeutic strategies not only for neuropathic pain but also for other CNS diseases.

# Materials and methods

## Key resources table

| Reagent type (species) or resource | Designation | Source or reference | Identifiers | Additional information |
|---|---|---|---|---|
| Strain, strain background (include species and sex here) | C57BL/6 J | The Jackson Laboratory Japan | NA | |
| Genetic reagent (include species here) | B6.Cg-Tg(Itgax-Venus)Mnz/J | Jackson Laboratory | RRID:IMSR_JAX:008829 | |
| Genetic reagent (include species here) | B6.FVB-1700016L21RikTg(Itgax-DTR/EGFP)57Lan/J | Jackson Laboratory | RRID:IMSR_JAX:004509 | |
| Genetic reagent (include species here) | B6.129P2(Cg)-Cx3cr1tm2.1(cre/ERT2)Litt/WganJ | Jackson Laboratory | RRID:IMSR_JAX:021160 | |
| Genetic reagent (include species here) | B6.129(FVB)-Igf1tm1Dlr/J | Jackson Laboratory | RRID:IMSR_JAX:016831 | |
| Antibody | Guinea pig anti-IBA1 | Synaptic Systems | RRID:AB_2924932 | 1:2000 |
| Antibody | Rabbit anti-P2Y12R | AnaSpec | RRID:AB_2298886 | 1:2000 |
| Antibody | Rabbit anti-GFP | MBL International | RRID:AB_591819 | 1:1000 |
| Antibody | APC anti-mouse CD169 | Biolegend | RRID:AB_2565640 | 1:500 |
| Antibody | Rabbit anti-CCR2 | Abcam | RRID:AB_2893307 | 1:500 |
| Antibody | Goat Anti-Guinea pig IgG H&L (Alexa Fluor 405) | Abcam | RRID:AB_2827755 | 1:1000 |
| Antibody | Goat anti-Rabbit IgG (H+L) Highly Cross-Adsorbed Secondary Antibody, Alexa Fluor 488 | Thermo Fisher Scientific | RRID:AB_2576217 | 1:1000 |
| Antibody | Goat anti-Rabbit IgG (H+L) Highly Cross-Adsorbed Secondary Antibody, Alexa Fluor 546 | Thermo Fisher Scientific | RRID:AB_2534093 | 1:1000 |
| Antibody | Rat Anti-CD16 /CD32 Monoclonal Antibody, Unconjugated, Clone 2.4G2 | BD Biosciences | RRID:AB_394657 | 1:200 |
| Antibody | Rat Anti-CD11b Monoclonal Antibody, Alexa Fluor 647 Conjugated, Clone M1/70 | BD Biosciences | RRID:AB_396796 | 1:1000 |
| Antibody | PE anti-mouse CD45 | BioLegend | RRID:AB_312971 | 1:1000 |

*Continued on next page*

*Continued*

| Reagent type (species) or resource | Designation | Source or reference | Identifiers | Additional information |
|---|---|---|---|---|
| Antibody | Brilliant Violet 785(TM) anti-mouse/human CD11b | BioLegend | RRID:AB_2561373 | 1:1000 |
| Antibody | APC anti-mouse CD206 (MMR) | BioLegend | RRID:AB_10900231 | 1:200 |
| Peptide, recombinant protein | Recombinant Murine IL-4 | PeproTech | 214–14 | |
| Commercial assay or kit | Myelin Removal Beads II | Miltenyi Biotec | 130-096-433 | |
| Commercial assay or kit | MACS LS column | Miltenyi Biotec | 130-042-401 | |
| Commercial assay or kit | Quick-RNA Micro-Prep kit | ZYMO | R1051 | |
| Commercial assay or kit | Trisure | Bioline | BIO-38032 | |
| Commercial assay or kit | Prime Script reverse transcriptase | Takara | 2680B | |
| Commercial assay or kit | Premix Ex Taq (Probe qPCR) | Takara | RR390B | |
| Chemical compound, drug | tamoxifen | SIGMA | T5648 | |
| Chemical compound, drug | Diphtheria Toxin Solution | Wako | 048–34371 | |
| Chemical compound, drug | collagenase D | Roche | 11088866001 | |
| Chemical compound, drug | dispase | GIBCO | 17105041 | |
| Chemical compound, drug | RNAlater | Invitrogen | AM7021 | |
| Chemical compound, drug | 7-aminoactinomycin D | Miltenyi Biotec | 170-081-088 | |
| Software, algorithm | LSM700 Imaging System | Carl Zeiss | NA | |
| Software, algorithm | CytoFLEX SRT | Beckman Coulter | NA | |
| Software, algorithm | FlowJo | BD | RRID:SCR_008520 | |
| Software, algorithm | FACSAria III | BD Biosciences | NA | |
| Software, algorithm | QuantStudio 3 | Applied Biosystems | NA | |
| Software, algorithm | Prism 7 | GraphPad | NA | |

## Mice

C57BL/6 mice were purchased from Charles River Japan (Kanagawa, Japan). *Itgax-Venus* mice (B6.Cg-Tg(*Itgax-Venus*)Mnz/J) (*Lindquist et al., 2004*), *Itgax-DTR-EGFP* mice (B6.FVB-1700016L21Rik$^{Tg(Itgax-DTR/EGFP)57Lan/J}$) (*Jung et al., 2002*), *Cx3cr1*$^{CreERT2/+}$ mice (B6.129P2(Cg)-*Cx3cr1*$^{tm2.1(cre/ERT2)Litt}$/WganJ) (*Parkhurst et al., 2013*), *Igf1*$^{flox/flox}$ mice (B6.129(FVB)-*Igf1*$^{tm1Dlr}$/J) *Liu et al., 1998* were purchased from Jackson Laboratory (Bar Harbor, ME). The data were obtained from male mice except for the data shown in *Figures 1D, 3 and 7C*. For induction of Cre recombinase, *Cx3cr1*$^{CreERT2/+}$; *Igf1*$^{flox/flox}$ mice were injected subcutaneously with 4 mg of tamoxifen (Sigma, St. Louis, MO) dissolved in 200 µL corn oil (Wako, Osaka, Japan) twice with a 48 hr interval. Four weeks later, when gene-deleting CX3CR1$^+$ cells in peripheral tissues including the DRG has been shown to be replaced by non-deleting cells (*Parkhurst et al., 2013*; *Peng et al., 2016*), the mice were subjected to nerve injury. All mice used were aged 5–14 weeks at the start of each experiment and were housed individually or in groups at a temperature of 22 ± 1°C with a 12 hr light–dark cycle and were fed food and water ad libitum. All animal experiments were conducted according to relevant national and international guidelines contained in the 'Act on Welfare and Management of Animals' (Ministry of Environment of Japan) and 'Regulation of Laboratory Animals' (Kyushu University) and under the protocols approved by the Institutional Animal Care and Use committee review panels at Kyushu University.

## Peripheral nerve injury

We used two injury models, spinal nerve transection (SpNT) (*Kohno et al., 2022*; *Masuda et al., 2017*), which is a modified spinal nerve injury model (*Ho Kim and Mo Chung, 1992*), and spared nerve injury (SNI) (*Decosterd and Woolf, 2000*). For the SpNT model, under isoflurane (2%) anesthesia, a

small incision at L3–S1 was made. The paraspinal muscle and fat were removed from the L5 transverse process, which exposed the parallel-lying L3 and L4 spinal nerves. The L4 nerve was then carefully isolated and cut while leaving the L3 spinal nerve intact. The wound and the surrounding skin were sutured with 5–0 silk. For the SNI model, according to a previously reported method (*Cichon et al., 2018*), under isoflurane (2%) anesthesia, the skin on the left thigh was open, and terminal branches of the sciatic nerve were exposed by separating the thigh muscle layer using scissors and blunt forceps. Two of the three branches, the tibial and common peroneal nerves, were tightly ligated with 7–0 Nylon (NM-01, WASHIESU MEDICAL), and a small portion of the nerves distal to the ligation was removed while leaving the sural nerve intact. The wound and the surrounding skin were sutured with 5–0 silk.

## Immunohistochemistry

According to our previously reported method (*Kohno et al., 2022*), mice were deeply anesthetized by i.p. injection of pentobarbital and perfused transcardially with phosphate buffered saline (PBS) followed by ice-cold 4% paraformaldehyde/PBS. Transverse L4 spinal cord sections (30 μm) or L4 dorsal root ganglion (DRG) sections (20 μm) were incubated for 48 hr at 4 °C with primary antibody for IBA1 (1:2000; Cat# 234 308, Synaptic Systems), P2Y12R (1:2000; Cat# AS-55043A, AnaSpec), CCR2 (1:500; Cat# ab273050, Abcam), APC-conjugated CD169 (1:500, Cat# 142417, Biolegend), PE-conjugated CD11b (1:1000, Cat# 101207, Biolegend), or GFP (1:1000; Cat# 598, MBL Life science). Tissue sections were incubated with secondary antibodies conjugated to Alexa Fluor 405 (1:1000, Abcam), 488, or 546 (1:1000, Molecular Probes) and mounted with Vectashield. Three to five sections from one tissue were randomly selected and analyzed using an LSM700 Imaging System (Carl Zeiss, Japan).

## Behavioral test

To assess mechanical sensitivity, calibrated von Frey filaments (0.02–2.0 g, North Coast Medical, USA) were applied to the plantar surfaces of the hindpaws of mice with or without PNI (*Kohno et al., 2022*) and the 50% PWT was determined using the up–down method (*Chaplan et al., 1994*).

## Intrathecal injection

Under 2% isoflurane anesthesia, a 30 G needle attached to a 25 μL Hamilton syringe was inserted into the intervertebral space between L5 and L6 spinal vertebrae in mice, as previously described (*Kohno et al., 2022*; *Hylden and Wilcox, 1980*). Recombinant Murine IL-4 was purchased from Peprotech (Cat# 214–14). IL-4 (40 ng/μL in PBS, 5 μL/mouse) or PBS as a control was intrathecally injected once a day for 3 days from day 14 post-PNI. DTX was purchased from Wako (Cat# 048–34371) and was injected intrathecally (0.1 ng/μL in PBS, 5 μL/mouse) or intraperitoneally (2 ng/g mouse) into *Itgax-DTR-EGFP* mice. Only mice whose hypersensitivity was attenuated by IL-4 treatment were used for the DTX-induced cell depletion experiment shown in *Figure 4C* (10 out of 12 mice were included) and *Figure 8B* (12 out of 15 mice were included).

## Flow cytometry

As previously described (*Kohno et al., 2022*), mice were deeply anesthetized by i.p. injection of pentobarbital and perfused transcardially with PBS to remove the circulating blood from the vasculature. The spinal cord was rapidly and carefully removed from the vertebral column and placed into ice-cold Hanks' balanced salt solution (HBSS). The 3rd–4th lumbar (L3/4) segments (2 mm long) of the spinal dorsal horn (SDH) ipsilateral and contralateral to nerve injury were separated. Unilateral spinal tissue pieces were treated with pre-warmed 0.8 mL enzymatic solution [0.2 U/mL collagenase D (Cat# 11088866001; Roche) and 4.3 U/mL dispase (Cat# 17105041; GIBCO)] in HBSS containing 2% fetal bovine serum (FBS) for 30 min at 37 °C. The tissues were homogenized by passing through a 23 G needle attached with a 1 mL syringe and were further incubated for 15 min at 37 °C. After that, the tissues were homogenized by passing twice through a 26 G needle, and the enzymatic reaction was stopped by adding EDTA (0.5 M). To count the number of CD11c$^+$ (Venus$^+$ or EGFP$^+$) microglia, cell suspension was blocked by incubating with Fc Block (Cat# 553142; BD Biosciences) for 5 min at 4 °C and immunostained with CD11b-AlexaFluor 647 (M1/70; 1:1000; Cat# 557686, BD Biosciences) and CD45-PE (1:1000; Cat# 103106, Biolegend) for 30 min at 4 °C in the dark. After washing, the pellet was resuspended in ice-cold HBSS-FBS and filtered through a 35 μm nylon cell strainer (BD Biosciences) to isolate tissue debris from the cell suspension. The total number of microglia (CD11b$^+$

CD45$^+$ cells) in the L3/4 spinal cord or SDH was analyzed using CytoFLEX SRT (Beckman Coulter) and FlowJo software (TreeStar). Microglia with Venus fluorescence higher in intensity than that observed in microglia of wild-type mice were analyzed as CD11c$^+$ microglia.

## Fluorescent-activated cell sorting (FACS)

Spinal tissues were removed and homogenized as described above. Myelin debris was removed from the cell suspension using Myelin Removal Beads II and a MACS LS column (Miltenyi Biotec, Bergisch-Gladbach, Germany) according to the manufacturer's protocol and our previously reported method (*Kohno et al., 2022*). After centrifugation (300 × *g*, 10 min, 4 °C), the cells were resuspended in HBSS containing 10% FBS. The cell suspension was blocked by incubating with Fc Block and immunostained with CD11b-Brilliant Violet 785 (1:1000; Cat# 101243, BioLegend), CD206-APC (1:200; Cat# 141708, BioLegend), and CD45-PE (1:1000; Cat# 103105, BioLegend) for 30 min at 4 °C in the dark. After washing, cell suspension was treated with 7-aminoactinomycin D (7-AAD; Miltenyi Biotec) and incubated for 10 min on ice for viability staining prior to cell sorting. CD11b$^+$ CD45$^+$ CD206$^{neg}$ 7-AAD$^{neg}$-singlet cells are gated as live microglia and sorted using FACSAria III (BD Biosciences) or CytoFLEX SRT. The sorted cells were directly collected in lysis buffer for total RNA extraction using the Quick-RNA Micro-Prep kit (ZYMO). CD11c$^{high}$ microglia were collected as cells with Venus fluorescence higher than the median Venus fluorescence of all CD11c$^+$ microglia (*Kohno et al., 2022*).

## mRNA extraction from spinal cord homogenates

Mice were deeply anesthetized by pentobarbital and perfused transcardially with ice-cold PBS followed by RNAlater Stabilization Solution (Cat# AM7021, Invitrogen). The isolated SDH tissue around the boundary between L3 and L4 segments was isolated and separated into the ipsilateral and contralateral sides, and the upper half portion of the SDH of each side was dissected as described previously (*Kohno et al., 2022*). Total RNA was extracted from tissue homogenates using Trisure (CAT# BIO-38032, Bioline, USA) according to manufacturer's protocol and purified with the Quick-RNA Micro-Prep kit (ZYMO).

## QPCR

As described previously (*Kohno et al., 2022*), the extracted RNA from the sorted cells or the tissue was transferred to reverse transcriptional reaction with Prime Script reverse transcriptase (2680B, Takara, Japan). Quantitative PCR (qPCR) was performed with Premix Ex Taq (Cat# RR390B, Takara, Japan) using QuantStudio 3 (Applied Biosystems, USA). Expression levels were normalized to the values for *Actb*. The sequences of TaqMan primer pairs and probe are described below: *Actb*: 5'-FAM-CCTG GCCTCACTGTCCACCTTCCA-TAMRA-3' (probe), 5'-CCTGAGCGCAAGTACTCTGTGT-3' (forward primer), 5'-CTGCTTGCTGATCCACATCTG-3' (reverse primer); *Igf1*: 5'- /56-FAM/TCCGGAAGC/ ZEN/AACACTCACATCCACAA/3IABkFQ/–3' (probe), 5'- AGTACATCTCCAGTCTCCTCA-3' (forward primer), 5'- ATGCTCTTCAGTTCGTGTGT-3' (reverse primer).

## Quantification and statistical analysis

All data are shown as mean ± s.e.m. Statistical significance was determined using the unpaired t-test, two-way ANOVA with post hoc Tukey's multiple comparison test or one-way ANOVA with post hoc Tukey's multiple comparison test using GraphPad Prism 7 software. Differences were considered significant at $p < 0.05$.

## Acknowledgements

We would like to thank Editage for editing a draft of this manuscript. This work was supported by Japan Society for the Promotion of Science (JSPS) KAKENHI Grants JP19H05658 (MT), JP20H05900 (MT) and JP24H00067 (MT), by the Core Research for Evolutional Science and Technology (CREST) program from AMED under Grant Number 25gm1510013h (MT) by Research Support Project for Life Science and Drug Discovery (Basis for Supporting Innovative Drug Discovery and Life Science Research (BINDS)) from AMED under Grant Number JP25ama121031 (MT). TM was supported by AMED (JP20gm6310016, JP21wm0425001, JP23gm1910004, JP23jf0126004, 24zf0127012), JSPS (KAKENHI JP21H02752, JP22H05062, JP25H01009), The Mitsubishi Foundation, Daiichi Sankyo Foundation of Life Science, Mochida Memorial Foundation for Medical and Pharmaceutical Research,

Astellas Foundation for Research on Metabolic disorders, Ono Pharmaceutical Foundation for Oncology, Immunology and Neurology, The Nakajima Foundation, The Uehara Memorial Foundation and Takeda Science Foundation. KK was supported by JSPS (KAKENHI JP23K19403, JP24K18621), JST (ACT-X Grant Number JPMJAX232B), Chugai Foundation for Innovative Drug Discovery Science, and The Ichiro Kanehara Foundation. This work was supported in part by the MEXT Cooperative Research Project Program, Medical Research Center Initiative for High Depth Omics, and CURE:-JPMXP1323015486 for MIB, Kyushu University.

## Additional information

### Funding

| Funder | Grant reference number | Author |
|---|---|---|
| Japan Society for the Promotion of Science | JP23K19403 | Keita Kohno |
| Japan Society for the Promotion of Science | JP24K18621 | Keita Kohno |
| Japan Science and Technology Agency | 10.52926/JPMJAX232B | Keita Kohno |
| Japan Agency for Medical Research and Development | JP20gm6310016 | Takahiro Masuda |
| Japan Agency for Medical Research and Development | JP21wm0425001 | Takahiro Masuda |
| Japan Agency for Medical Research and Development | JP23gm1910004 | Takahiro Masuda |
| Japan Agency for Medical Research and Development | JP23jf0126004 | Takahiro Masuda |
| Japan Agency for Medical Research and Development | 24zf0127012 | Takahiro Masuda |
| Japan Society for the Promotion of Science | JP21H02752 | Takahiro Masuda |
| Japan Society for the Promotion of Science | JP22H05062 | Takahiro Masuda |
| Japan Society for the Promotion of Science | JP25H01009 | Takahiro Masuda |
| Daiichi Sankyo Foundation of Life Science | | Takahiro Masuda |
| Mochida Memorial Foundation for Medical and Pharmaceutical Research | | Takahiro Masuda |
| Astellas Foundation for Research on Metabolic Disorders | | Takahiro Masuda |
| Ono Pharmaceutical Foundation for Oncology, Immunology and Neurology | | Takahiro Masuda |
| The Nakajima Foundation | | Takahiro Masuda |

| Funder | Grant reference number | Author |
|---|---|---|
| Uehara Memorial Foundation | | Takahiro Masuda |
| Takeda Science Foundation | | Takahiro Masuda |
| Japan Society for the Promotion of Science | JP19H05658 | Makoto Tsuda |
| Japan Society for the Promotion of Science | JP20H05900 | Makoto Tsuda |
| Japan Society for the Promotion of Science | JP24H00067 | Makoto Tsuda |
| Japan Agency for Medical Research and Development | 25gm1510013h | Makoto Tsuda |
| Japan Agency for Medical Research and Development | JP25ama121031 | Makoto Tsuda |

The funders had no role in study design, data collection and interpretation, or the decision to submit the work for publication.

## Author contributions

Keita Kohno, Funding acquisition, Investigation, Visualization, Writing – original draft, Writing – review and editing; Ryoji Shirasaka, Keita Hirose, Investigation; Takahiro Masuda, Supervision, Funding acquisition; Makoto Tsuda, Conceptualization, Supervision, Funding acquisition, Writing – original draft, Project administration, Writing – review and editing

## Author ORCIDs

Keita Kohno https://orcid.org/0000-0001-5038-5680
Takahiro Masuda https://orcid.org/0000-0003-3687-452X
Makoto Tsuda https://orcid.org/0000-0003-0585-9570

## Ethics

All animal experiments were conducted according to relevant national and international guidelines contained in the 'Act on Welfare and Management of Animals' (Ministry of Environment of Japan) and 'Regulation of Laboratory Animals' (Kyushu University) and under the protocols approved by the Institutional Animal Care and Use committee review panels at Kyushu University (A24-305-1, A24-466-0).

Reviewer #2 (Public review): https://doi.org/10.7554/eLife.105087.3.sa1
Author response https://doi.org/10.7554/eLife.105087.3.sa2

---

# Additional files

## Supplementary files

MDAR checklist

## Data availability

All data generated or analysed during this study are included in the manuscript and supporting files.

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
