## [Editor Report · eLife Assessment]

This study is **important** as it highlighted how IL-4 regulates the reactive state of a specific microglial population by increasing the proportion of CD11c+ microglial cells and ultimately suppressing neuropathic pain. The study employs a combination of behavioral assays, pharmacogenetic manipulation of microglial populations, and characterization of microglial markers to address these questions. It provided **convincing** evidence for the proposed mechanism of IL-4-mediated microglial regulation in neuropathic pain.

---

## [Referee Report · Reviewer #2 (Public review)]

Summary:

The authors aimed to investigate how IL-4 modulates the reactive state of microglia in the context of neuropathic pain. Specifically, they sought to determine whether IL-4 drives an increase in CD11c+ microglial cells, a population associated with anti-inflammatory responses, and whether this change is linked to the suppression of neuropathic pain. The study employs a combination of behavioral assays, pharmacogenetic manipulation of microglial populations, and characterization of microglial markers to address these questions.

Strengths:

Strengths: The methodological approach in this study is robust, providing convincing evidence for the proposed mechanism of IL-4-mediated microglial regulation in neuropathic pain. The experimental design is well thought out, utilizing two distinct neuropathic pain models (SpNT and SNI), each yielding different outcomes. The SpNT model demonstrates spontaneous pain remission and an increase in the CD11c+ microglial population, which correlates with pain suppression. In contrast, the SNI model, which does not show spontaneous pain remission, lacks a significant increase in CD11c+ microglia, underscoring the specificity of the observed phenomenon. This design effectively highlights the role of the CD11c+ microglial population in pain modulation. The use of behavioral tests provides a clear functional assessment of IL-4 manipulation, and pharmacogenetic tools allow for precise control of microglial populations, minimizing off-target effects. Notably, the manipulation targets the CD11c promoter, which presumably reduces the risk of non-specific ablation of other microglial populations, strengthening the experimental precision. Moreover, the thorough characterization of microglial markers adds depth to the analysis, ensuring that the changes in microglial populations are accurately linked to the behavioral outcomes.

Weaknesses:

One potential limitation of the study is that the mechanistic details of how IL-4 induces the observed shift in microglial populations are not fully explored. While the study demonstrates a correlation between IL-4 and CD11c+ microglial cells, a deeper investigation into the specific signaling pathways and molecular processes driving this population shift would greatly strengthen the conclusions. Additionally, the paper does not clearly integrate the findings into the broader context of microglial reactive state regulation in neuropathic pain.

Comments on revisions:

In the revised manuscript, the authors have successfully addressed my previous concerns as well as the other reviewers. I do not have further concerns about this study.

---

## [Author Response]

The following is the authors’ response to the original reviews.

**Reviewer #1 (Public review):**
Summary:Kohno et al. examined whether the anti-inflammatory cytokine IL-4 attenuates neuropathic pain by promoting the emergence of antinociceptive microglia in the dorsal horn of the spinal cord. In two models of neuropathic pain following peripheral nerve injury, intrathecal administration of IL-4 once a day for 3 days from day 14 to day 17 after injury, attenuates hypersensitivity to mechanical stimuli in the hind paw ipsilateral to nerve injury. Such an antinociceptive effect correlates with a higher number of CD11c+microglia in the dorsal horn of the spinal cord which is the termination area for primary afferent fibres injured in the periphery. Interestingly, CD11c+ microglia emerge spontaneously in the dorsal horn in concomitance with the resolution of pain in the spinal nerve model of pain, but not in the spared nerve injury model where pain does not resolve, confirming that this cluster of microglia is involved in resolution pain.Based on existing evidence that the receptor for IL-4, namely IL-4R, is expressed by microglia, the authors suggest that IL-4R mediates IL-4 effect in microglia including up-regulation of Igf1 mRNA. They have previously reported that IGF-1 can attenuate pain neuron activity in the spinal cord.Strengths:This study includes cutting-edge techniques such as flow cytometry analysis of microglia and transgenic mouse models.Weaknesses:The conclusion of this paper is supported by data, but the interpretation of some data requires clarification.

We appreciate the reviewer's careful reading of our paper. According to the reviewer's comments, we have performed new immunohistochemical experiments and added some discussion in the revised manuscript (please see the point-by-point responses below).

**Reviewer #2 (Public review):**
Summary:The authors aimed to investigate how IL-4 modulates the reactive state of microglia in the context of neuropathic pain. Specifically, they sought to determine whether IL-4 drives an increase in CD11c+ microglial cells, a population associated with anti-inflammatory responses and whether this change is linked to the suppression of neuropathic pain. The study employs a combination of behavioral assays, pharmacogenetic manipulation of microglial populations, and characterization of microglial markers to address these questions.Strengths:The methodological approach in this study is robust, providing convincing evidence for the proposed mechanism of IL-4-mediated microglial regulation in neuropathic pain. The experimental design is well thought out, utilizing two distinct neuropathic pain models (SpNT and SNI), each yielding different outcomes. The SpNT model demonstrates spontaneous pain remission and an increase in the CD11c+ microglial population, which correlates with pain suppression. In contrast, the SNI model, which does not show spontaneous pain remission, lacks a significant increase in CD11c+ microglia, underscoring the specificity of the observed phenomenon. This design effectively highlights the role of the CD11c+ microglial population in pain modulation. The use of behavioral tests provides a clear functional assessment of IL-4 manipulation, and pharmacogenetic tools allow for precise control of microglial populations, minimizing off-target effects. Notably, the manipulation targets the CD11c promoter, which presumably reduces the risk of non-specific ablation of other microglial populations, strengthening the experimental precision. Moreover, the thorough characterization of microglial markers adds depth to the analysis, ensuring that the changes in microglial populations are accurately linked to the behavioral outcomes.Weaknesses:One potential limitation of the study is that the mechanistic details of how IL-4 induces the observed shift in microglial populations are not fully explored. While the study demonstrates a correlation between IL-4 and CD11c+ microglial cells, a deeper investigation into the specific signaling pathways and molecular processes driving this population shift would greatly strengthen the conclusions. Additionally, the paper does not clearly integrate the findings into the broader context of microglial reactive state regulation in neuropathic pain.

We thank the reviewer for these insightful comments on our paper. As the reviewer's suggested, further investigation of the specific signaling pathways and molecular processes by which IL-4 induces a transition of spinal microglia to the CD11c+ state would strengthen our conclusion and also provide important clues to discovering new therapeutic targets. In revising the manuscript, we have included this in the Discussion section (line 264-267), and we hope that future studies clarify these points. As for the additional comment, we have added a brief summary of existing research on microglial function in neuropathic pain at the beginning of the Discussion section (line 188–196).

**Reviewer #1 (Recommendations for the authors):**
The conclusions of this paper are supported by data, but the interpretation of some data requires clarification.(1) In Figure 1D and Figure 7 C, CD11c+ microglia numbers are higher in contralateral dorsal horns after IL-4 administration despite IL-4 having no effect on pain thresholds. The authors should discuss these findings.

As the reviewer pointed out, IL-4 increased the number of CD11c^+^ microglia in the contralateral spinal dorsal horn (SDH) but did not affect pain thresholds in the contralateral hindpaw. The data seem to be related to the selective effect of CD11c+ microglia and their factors (especially IGF1) on nerve injury-induced pain hypersensitivity. In fact, depletion of CD11c+ spinal microglia and intrathecal administration of IGF1 do not elevate pain threshold of the contralateral hindpaw (Science 376: 86–90, 2022). We have added above statement in the Discussion section (line 208– 213).

(2) Do monocytes infiltrate the dorsal horn and DRG after intrathecal injections?

To address this reviewer's comment, we performed new immunohistochemical experiments to analyze monocytes in the SDH using an antibody for CD169 (a marker for bone marrow-derived monocytes/macrophages, but not for resident microglia) (J Clin Invest 122: 3063– 3087, 2012; Cell Rep 3: 605–614, 2016) and found no CD169+ monocytes in the SDH parenchyma after SpNT. Consistent with this data, we have previously demonstrated that few bone marrow-derived monocytes/macrophages are recruited to the SDH after SpNT (Sci Rep 6: 23701, 2016). Similarly, no CD169+ monocytes in the SDH parenchyma were observed in SpNT mice treated intrathecally with PBS or IL-4 (Figure 1—figure supplement 1A).

In the DRG, CD169 is constitutively expressed in macrophages. Thus, in accordance with a recent report showing that monocytes infiltrating the DRG are positive for chemokine (C-C motif) receptor 2 (CCR2) (J Exp Med 221: e20230675, 2024), we analyzed CCR2+ cells and found that CCR2+ IBA1dim monocytes were observed in the capsule and parenchyma of the DRG of naive mice (Figure 1—figure supplement 1B). After SpNT, CCR2+ IBA1dim monocytes in the DRG parenchyma increased. Intrathecal treatment of IL-4 increased CCR2+ IBA1dim cells in the DRG capsule. However, the involvement of these monocytes in the DRG in IL-4-induced alleviation of neuropathic pain is unclear and warrants further investigation. In revising the manuscript, we have included additional data (Figure 1—figure supplement 1) and corresponding text in the Results (line 112–114) and Discussion section (line 218–222).

(3) In Figure 4, depletion of CD11c+ cells in dorsal root ganglia (DRG) ameliorates neuropathic thresholds but does not alter the anti-nociceptive effect of IL-4 injected intrathecal. It appears that CD11c+ macrophages in DRG have an opposite role to CD11c+ microglia in the spinal cord. Please discuss this result.

We apologize for the confusion. The aim of the experiments in Figure 4 was to examine the contribution of CD11c+ cells in the DRG to the pain-alleviating effect of intrathecal IL-4. For this aim, we depleted CD11c+ cells in the DRG (but not in the SDH) by intraperitoneal injection of diphtheria toxin (DTX) immediately after the behavioral measurements performed on day 17 (Fig. 4A, B). On day 18, the paw withdrawal threshold of DTX-treated mice was almost similar to that of PBS-treated mice, indicating that the depletion of CD11c+ cells in the DRG does not affect the pain-alleviating effect of IL-4. These data are in stark contrast to those obtained from mice with depletion of CD11c+ cells in the SDH by intrathecal DTX (the depletion canceled the IL-4's effect) (Figure 2A). Thus, it is conceivable that CD11c+ cells in the DRG are not involved in the IL-4-induced alleviating effect on neuropathic pain. Because the confusion might be related to the statement in this paragraph of the initial version, we thus modified our statements to make this point more clearly (line 133–139).

**Reviewer #2 (Recommendations for the authors):**
A discussion addressing how these results fit into existing research on microglial function in pain would enhance the study's impact.

A brief summary of existing research on microglial function in neuropathic pain has been included at the beginning of the Discussion section (line 188–196).

It would be helpful for the authors to elaborate on the implications of their findings within the larger landscape of immune regulation in neuropathic pain.

Our present findings showed an ability of IL-4, known as a T-cell-derived factor, to increase CD11c+ microglia and to control neuropathic pain. Furthermore, recent studies have also indicated that immune cells such as CD8+ T cells infiltrating into the spinal cord (Neuron 113: 896-911.e9, 2025), and regulatory T cells (eLife 10: e69056, 2021; Science 388: 96–104, 2025) and MRC1+ macrophages in the spinal meninges (Neuron 109: 1274–1282, 2021) have important roles in regulating microglial states and neuropathic pain. Thus, these findings provide new insights into the mechanisms of the neuro-immune interactions that regulate neuropathic pain. In revising the manuscript, we have added above statement in the Discussion section (line 254–260).

Furthermore, a discussion on how these findings could inform the development of targeted therapies that modulate microglial populations in a controlled, disease-specific manner would be valuable. Exploring how these insights could lead to novel treatment strategies for neuropathic pain could provide important future directions for the research and broader clinical applications.

We appreciate the reviewer's valuable suggestion. Our current data, demonstrating that IL-4 increases CD11c+ microglia without affecting the total number of microglia, could open a new avenue for developing strategies to modulate microglial subpopulations through molecular targeting, which may lead to new analgesics. However, given IL-4's association with allergic responses, targeting microglia-selective molecules involved in shifting microglia toward the CD11c+ state—such as intracellular signaling molecules downstream of IL-4 receptors—may offer a more selective and safer therapeutic approach. Moreover, since CD11c+ microglia have been implicated in other CNS diseases [e.g., Alzheimer disease (Cell 169: 1276–1290, 2017), amyotrophic lateral sclerosis (Nat Neurosci 25: 26–38, 2022), and multiple sclerosis (Front Cell Neurosci 12: 523, 2019)], further investigations into the mechanisms driving CD11c+ microglia induction could provide insights into novel therapeutic strategies not only for neuropathic pain but also for other CNS diseases. In revising the manuscript, we have added above statement in the Discussion section (line 260–271).